# *Cdh5*-lineage–independent origin of dermal lymphatics shown by temporally restricted lineage tracing

Yan Zhang*, Henrik Ortsäter* , Ines Martinez-Corral*, Taija Mäkinen

The developmental origins of lymphatic endothelial cells (LECs) have been under intense research after a century-long debate. Although previously thought to be of solely venous endothelial origin, additional sources of LECs were recently identified in multiple tissues in mice. Here, we investigated the regional differences in the origin(s) of the dermal lymphatic vasculature by lineage tracing using the pan-endothelial *Cdh5-CreER^{T2}* line. Tamoxifen-induced labeling of blood ECs at E9.5, before initiation of lymphatic development, traced most of the dermal LECs but with lower efficiency in the lumbar compared with the cervical skin. By contrast, when used at E9.5 but not at E11.5, 4-hydroxytamoxifen, the active metabolite of tamoxifen that provides a tighter window of Cre activity, revealed low labeling frequency of LECs, and lymphvasculogenic clusters in the lumbar skin in particular. Temporally restricted lineage tracing thus reveals contribution of LECs of *Cdh5*-lineage–independent origin to dermal lymphatic vasculature. Our results further highlight Cre induction strategy as a critical parameter in defining the temporal window for stage-specific lineage tracing during early developmental stages of rapid tissue differentiation.

## Introduction

Lymphatic vasculature was traditionally considered a mere drainage and transport system for excess tissue fluid and immune cells. Recent research has demonstrated that lymphatic vessels play an active role in many important physiological processes and common diseases, which has raised interest in better understanding the underlying mechanisms (Oliver et al, 2020; Petrova & Koh, 2020).

During development, lymphatic endothelial cells (LECs) emerge initially through transdifferentiation of venous ECs that subsequently exit the veins to form the first lymphatic vessels (Wigle & Oliver, 1999). Inducible genetic lineage tracing using the *Prox1-*

*CreER^{T2}*, which is active in the differentiated LECs, further suggested a sole venous origin of the mammalian lymphatic vasculature (Srinivasan et al, 2007). However, recent studies demonstrated additional, including non-venous, sources of LECs in multiple organs, such as the mesentery (Stanczuk et al, 2015), the skin (Martinez-Corral et al, 2015; Pichol-Thievend et al, 2018), and the heart (Klotz et al, 2015; Gancz et al, 2019; Maruyama et al, 2019; Stone & Stainier, 2019; Lioux et al, 2020). The existence of tissue-specific LEC progenitors raises a question on their potential phenotypic and functional differences related to distinct developmental origins. In some cases, the precise cellular origin however remains unclear or controversial, which is in part because of limitations in lineage tracing technologies (Gutierrez-Miranda & Yaniv, 2020; Jafree et al, 2021).

For example, conflicting data exist on the origin(s) of dermal lymphatic vessels that exhibit region-specific temporal and morphogenetic differences in the development. In mouse embryos, dermal lymphatic-vessel formation is first observed in the cervical skin at around embryonic day (E) 12, and it occurs through lateral sprouting. In contrast, vessels in the lumbar skin and in the dorsal midline form through lymphvasculogenic assembly from progenitors starting from E12.5-E13 (Martinez-Corral et al, 2015; Pichol-Thievend et al, 2018). Genetic lineage tracing using the *Tie2-Cre* line, which efficiently labeled the blood endothelium, demonstrated contribution of non-*Tie2*–lineage (i.e., presumably non-endothelial) cells selectively into the lymphvasculogenic clusters (Martinez-Corral et al, 2015). Another study concluded that the isolated LEC clusters are endothelial-derived and form from the dermal blood capillary plexus through transdifferentiation (Pichol-Thievend et al, 2018). This conclusion was based on the analysis of several genetic models, most importantly temporally controlled tracing of *Cdh5*-lineage cells (Pichol-Thievend et al, 2018). Unlike in the earlier study (Martinez-Corral et al, 2015), incomplete *Tie2-Cre*–mediated recombination was observed and proposed to explain lack of LEC labeling (Pichol-Thievend et al, 2018). The discrepancy in the recombination efficiency may be explained by the use of different transgenic *Tie2-Cre* lines (Kisanuki et al, 2001; Koni et al, 2001). In summary, the existing literature thus provides evidence for both

---

Department of Immunology, Genetics and Pathology, Uppsala University, Uppsala, Sweden

Correspondence: taija.makinen@igp.uu.se
Ines Martinez-Corral's present address is University of Lille, Inserm, CHU Lille, Lille Neuroscience and Cognition, Laboratory of Development and Plasticity of the Neuroendocrine Brain, UMR-S, Labex DistAlz, Lille, France.
*Yan Zhang, Henrik Ortsäter, and Ines Martinez-Corral contributed equally to this work.

---

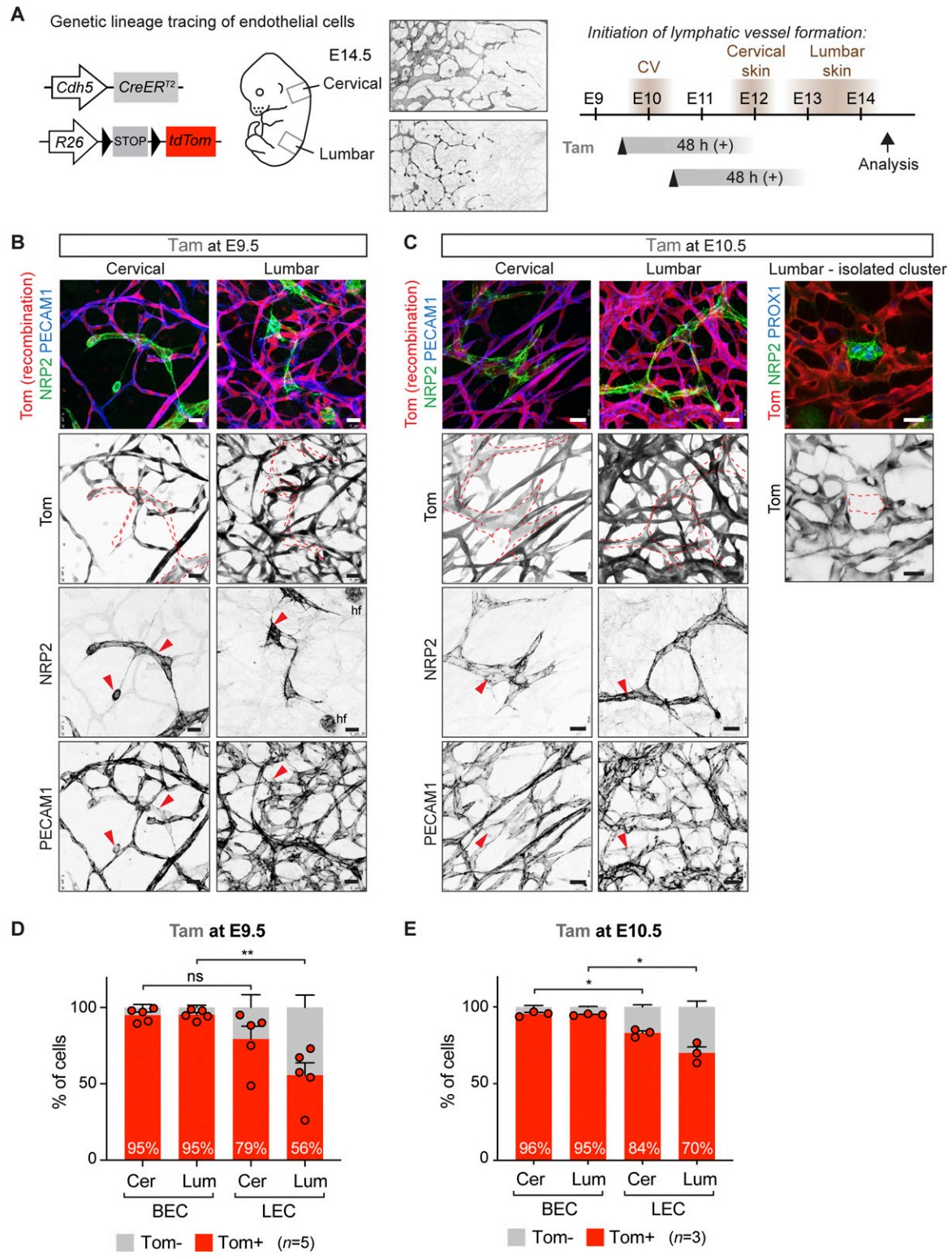

**Figure 1. Tamoxifen-induced lineage labeling of BECs before initiation of lymphatic development.**
**(A)** On the left: schematic of the *Cdh5-CreER$^{T2}$* transgene and *R26-tdTom* reporter for lineage tracing of BEC-derived lymphatic endothelial cells (LECs). On the right: timing of LEC differentiation from the cardinal vein (CV) and dermal lymphatic-vessel formation in the cervical and lumbar regions of the skin (shown by NRP2 whole-mount immunofluorescence), as well as the time points of tamoxifen administration and analysis are indicated. The estimated time window of Cre activity is up to at least 48 h. **(B, C)** Whole-mount immunofluorescence of E14.5 skin from *R26-tdTom;Cdh5-CreER$^{T2}$* embryos treated with tamoxifen at E9.5 (B) or at E10.5 (C). Dotted lines highlight lymphatic vessels and LEC clusters, defined by high NRP2 and low PECAM1 expression (red arrowheads). hf, hair follicle cells. **(D, E)** FACS analysis of ECs from *R26-tdTom;Cdh5-CreER$^{T2}$* embryos treated with tamoxifen at E9.5 (D) or at E10.5 (E). Note efficient Cre-mediated recombination (Tomato expression) in BECs and lower

non-endothelial (specifically, non-*Tie2*–lineage) and endothelial origins of the lymphvasculogenic dermal LECs.

Although Cre/loxP–based lineage tracing is a powerful method for the identification of all progeny produced by a single cell or groups of cells, it has certain notable limitations (Jensen & Dymecki, 2014; Martinez-Corral & Makinen, 2018). For example, constitutive Cre lines do not allow determining temporal lineage contributions and cannot distinguish de novo expression of Cre from a recombination event at an earlier developmental stage. Although they can be powerful when excluding the contribution of a particular cell lineage to a tissue of interest, incomplete Cre-mediated recombination in the target cell population has to be carefully assessed. Importantly, the efficiency of recombination depends on the sensitivity of the Cre reporter line used (Álvarez-Aznar et al, 2020), which may critically affect the outcome of the tracing experiment. Inducible approaches using the CreER$^{T2}$ provide the advantage of transiently activating Cre at a specific developmental time point upon administration of tamoxifen or its active metabolite 4-hydroxytamoxifen (4-OHT). Both compounds have been used widely, and often interchangeably, in lineage tracing studies. There are however important differences related to their activity that profoundly affect the timing and length of the time window when they can induce recombination (Jensen & Dymecki, 2014). Despite being an often-crucial parameter, especially during early developmental stages of rapid growth, surprisingly little attention is given to the experimental conditions for induction of Cre activity.

Taking the above considerations into account, we attempted to address the controversy regarding the cellular origin(s) of the dermal lymphatic vasculature in mice. We show that temporally restricted genetic lineage tracing and analysis of different regions of the skin that develop through distinct morphogenetic processes and at different developmental stages reveals a significant *Cdh5*-lineage–independent origin of dermal lymphatic vessels, in particular in the lumbar region of the skin.

# Results and Discussion

### Tamoxifen-induced lineage labeling of BECs before initiation of lymphatic development results in partial tracing of dermal LECs

To re-examine the contributions of LECs of the proposed venous/blood capillary (Pichol-Thievend et al, 2018; Stone & Stainier, 2019) versus non-venous (Martinez-Corral et al, 2015) origins into dermal lymphatic vasculature, we used the *Cdh5-CreER$^{T2}$* line (Wang et al, 2010) that allows targeting and tracing the EC lineage at defined developmental stages (Fig 1A). Mice were crossed with the *R26-tdTom* reporter that expresses a red fluorescent protein variant tdTomato upon Cre activation. The *R26-tdTom* line was used previously to study LEC origins (Pichol-Thievend et al, 2018) and chosen because of its high sensitivity to report Cre activity (Álvarez-Aznar et al, 2020). Because of regional differences in the timing and the

morphogenetic process of dermal lymphatic-vessel development (Martinez-Corral et al, 2015), we analyzed the lineage labeled cells separately in cervical and lumbar regions of the skin (Fig 1A).

We first administered 1 mg of tamoxifen at E9.5 to induce Cre-mediated labeling of BECs before initiation of lymphatic-vessel development, thereby attempting to specifically target blood vessel–derived LECs. In agreement with the previous study (Pichol-Thievend et al, 2018), whole-mount immunofluorescence analysis of cervical skin of an E14.5 embryo revealed efficient labeling of both BECs and LECs, including isolated clusters of LECs at the dorsal midline (Fig 1B). Analysis of lumbar skin demonstrated a mixed population of Tomato$^+$ and Tomato$^-$ LECs (Fig 1B). Administration of tamoxifen one day later, at E10.5, led to an apparently higher frequency of Tomato$^+$ LECs (Fig 1C). Yet, Tomato$^-$ LECs and even entire clusters of LECs were frequently observed in the lumbar region of the skin (Fig 1C), suggesting *Cdh5*-independent origin of these cells.

### Selective underrepresentation of *Cdh5*-lineage LECs in the lumbar skin

Next, we used flow cytometry to quantitatively determine the frequency of Tomato-labeled BECs and LECs in the two regions of the skin. Live CD45$^-$CD11b$^-$ cells were separated into PECAM1$^+$PDPN$^-$ BECs and PECAM1$^+$PDPN$^+$ LECs and analyzed for Tomato expression (Fig S1A and B). Both mature lymphatic vessels and isolated LEC clusters were identified as PECAM1$^+$ (Fig 1B and C). FACS analysis confirmed efficient labeling of BECs in both cervical and lumbar regions of the skin (Fig 1D). With the exception of one embryo, LECs in the cervical region of the skin also showed efficient labeling (79.4% ± 8.4% [*n* = 5], Fig 1D), reproducing the labeling efficiency of LEC clusters at the dermal midline (74%) observed in the previous study (Pichol-Thievend et al, 2018). However, in line with the immunofluorescence data (Fig 1B), the labeling frequency was significantly reduced in the lumbar region of the skin where only 55.6% ± 8.1% (*n* = 5) of LECs were Tomato$^+$ (Fig 1D). Three independent litters were analyzed to exclude inter-litter variations in the stage of embryos that may affect *Cdh5-CreER$^{T2}$*-mediated recombination efficiency depending on the developmental state of the vasculature. Analysis of untreated litters of E14.5 *R26-tdTom;Cdh5-CreER$^{T2}$* embryos did not show significant tamoxifen-independent recombination in BECs (0.9% ± 0.5% [*n* = 4]) or LECs (0.7% ± 0.3% [*n* = 4]).

Compared with embryos treated at E9.5, those treated at E10.5 showed a similar high recombination frequency of BECs (95–96%) and LECs of the cervical region (79% at E9.5 versus 84% at E10.5) (Fig 1D and E). In contrast, recombination of lumbar LECs was increased (56% at E9.5 versus 70% at E10.5) (Fig 1D and E), consistent with more efficient targeting of later differentiating LECs in this region with a later induction time point.

Taken together, and in agreement with previous data (Pichol-Thievend et al, 2018), tamoxifen-induced *Cdh5-CreER$^{T2}$*–mediated lineage labeling of BECs at E9.5, before initiation of lymphatic

---

recombination in LECs in the lumbar region of skin. Cer, cervical; Lum, lumbar. **(D, E)** Data represent mean % of Tomato$^+$ cells (*n* = 5 embryos from 3 liters (D) or *n* = 3 embryos from 1 liter (E), dots indicate individual embryos) ± SEM. *P*-value, paired two-tailed *t* test. \**P* < 0.05, \*\**P* < 0.01. Scale bars: 25 *μ*m.
Source data are available for this figure.

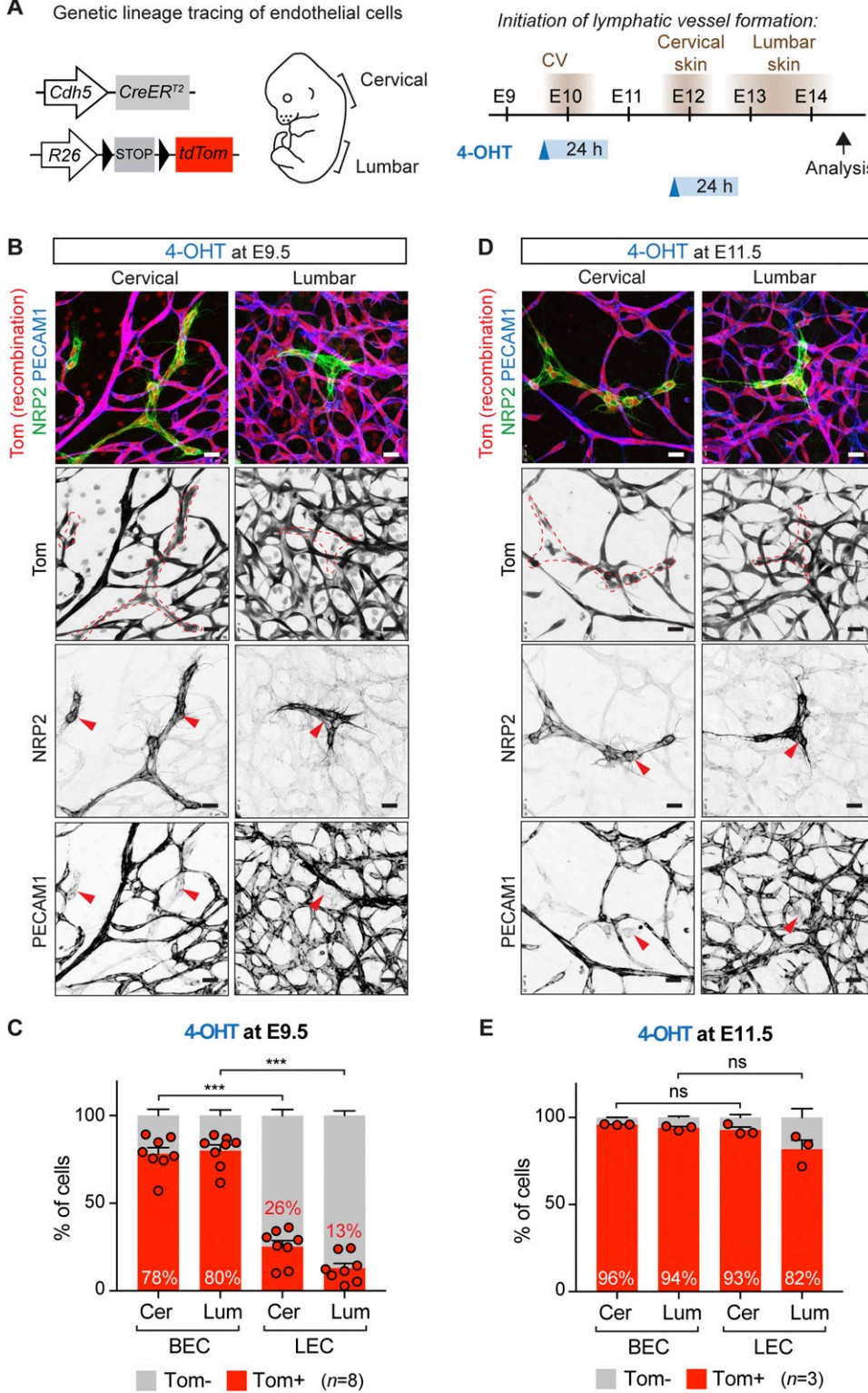

**Figure 2. Temporally restricted lineage tracing of BEC-derived lymphatic endothelial cells (LECs) using 4-OHT.**
**(A)** On the left: schematic of the *Cdh5-CreER$^{T2}$* transgene and *R26-tdTom* reporter for lineage tracing of BEC-derived LECs. On the right: timing of LEC differentiation from the cardinal vein (CV) and dermal lymphatic-vessel formation in the cervical and lumbar regions of the skin (depicted on the left), as well as the time points of 4-OHT administration and analysis are indicated. The estimated time window of Cre activity is up to 24 h. **(B, D)** Whole-mount immunofluorescence of E14.5 skin from *R26-tdTom;Cdh5-CreER$^{T2}$* embryos treated with 4-OHT at E9.5 (B) or at E11.5 (D). Dotted lines highlight lymphatic vessels and LEC clusters, defined by high NRP2 and low PECAM1 expression (red arrowheads). **(C, E)** FACS analysis of ECs from *R26-tdTom;Cdh5-CreER$^{T2}$* embryos treated with 4-OHT at E9.5 (C) or at E11.5 (E). Note efficient Cre-mediated recombination (Tomato expression) in BECs but low recombination in LECs in embryos treated at E9.5. Cer, cervical; Lum, lumbar. **(C, E)** Data represent mean % of Tomato⁺ cells ($n$ = 8 embryos from 3 liters (C) or $n$ = 3 embryos from 1 liter (E), dots indicate individual embryos) ± SEM. *P*-value, paired two-tailed *t* test. ***$P$ < 0.001. Scale bars: 25 µm. Source data are available for this figure.

development, results in tracing of dermal LECs. However, quantitative analysis of recombination frequency in the different skin regions revealed an underrepresentation of Tomato⁺ LECs in the lumbar skin that cannot be explained by lower labeling efficiency of BECs in the same region. Thus, our data argue against sole *Cdh5*-lineage venous and blood capillary origin of dermal LECs and instead provide additional evidence for their non-endothelial (or non–*Cdh5/Tie2* lineage) origin.

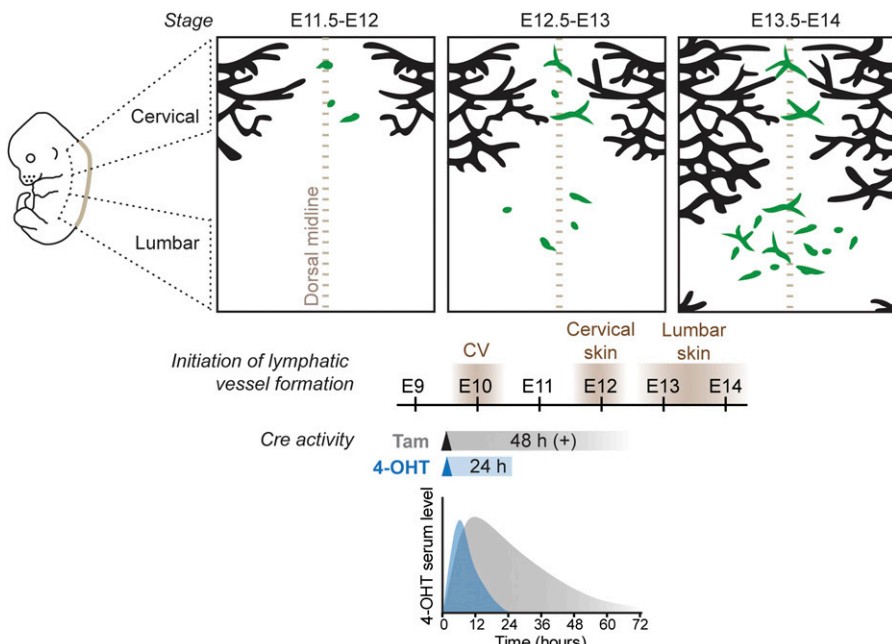

**Figure 3. Stages of dermal lymphatic-vessel development in relation to the window of Cre activity induced by tamoxifen or 4-OHT to trace BEC-derived lymphatic endothelial cells (LECs).** Schematic model of dermal lymphatic vascular development depicting sprouting lymphangiogenesis (black) in the lateral skin and lymphvasculogenic assembly of vessels from progenitors (green) in the dorsal midline of the skin (brown dotted line). Timing of LEC differentiation from the cardinal vein (CV) as well as lymphatic-vessel formation in the cervical and lumbar regions of the skin is indicated below. Tracing of CV-derived LECs by tam or 4-OHT–induced recombination results in different developmental windows of Cre activity, based on measurement of 4-OHT serum levels after a single intraperitoneal injection of tam or 4-OHT to mice (schematic graph based on Martinez-Corral et al [2015]).

## Temporally restricted genetic tracing using 4-OHT reveals *Cdh5*-lineage–independent origin of dermal LECs

Notably, the relatively long time window of Cre activity induced by tamoxifen administration, in relation to the rapid and region-specific process of dermal vascular development, raises an important question of the specificity of lineage labeling with this compound. High serum levels of the active metabolite 4-OHT were measured up to at least 48 h after tamoxifen administration (Zovein et al, 2008; Martinez-Corral et al, 2015). If an extended time window of Cre activity overlaps with the initiation of LEC differentiation (Fig 1A), the induction of Tomato expression may represent de novo *Cdh5*-driven Cre expression rather than demonstrating *Cdh5*-lineage BEC origin of these cells.

To allow tighter control of the window of Cre activity, we repeated the lineage tracing analysis using 4-OHT that is cleared from the systemic circulation within 24 h after administration (Martinez-Corral et al, 2015) (Fig 2A). Administration of 1 mg of 4-OHT at E9.5 led to partial labeling of LECs in the cervical skin despite efficient labeling of the blood vasculature (Fig 2B). In contrast, *Cdh5*-lineage Tomato⁺ LECs were rarely detected in the lumbar skin (Fig 2B). FACS analysis of dermal-cell suspensions of eight embryos from three different litters confirmed low LEC labeling in both skin regions (Fig 2C). Although labeling of BECs was also reduced compared with embryos that were treated with tamoxifen, this was not proportional to the remarkable decrease in the number of Tomato⁺ LEC that was even more pronounced in the lumbar region of the skin (Fig 2C).

These results suggest that a significant proportion of dermal LECs, in particular in the lumbar region of the skin, originate from non-*Cdh5*–lineage cells. To test if an extended window of Cre activity after tamoxifen administration at E9.5 could indeed explain partial labeling of dermal LECs through directly targeting

them at around E12, we administered 4-OHT at E11.5 (Fig 2A). Analysis by whole-mount immunofluorescence (Fig 2D) and flow cytometry (Fig 2E) showed efficient targeting of both BECs and LECs in both skin regions. This indicates that if Cre is active between E11.5-E12.5, differentiating LECs can be targeted, because of induction of *Cdh5* (encoding VE-cadherin) expression in these cells.

Taken together, our results provide important insight into the cellular origins of LECs that have after a century-long debate been under intense research (Gutierrez-Miranda & Yaniv, 2020). The previously accepted dogma that all mammalian LECs are of solely venous origin (Srinivasan et al, 2007) has been challenged recently. Lymphatic vasculature in the skin (Martinez-Corral et al, 2015; Pichol-Thievend et al, 2018) (Fig 3), mesentery (Stanczuk et al, 2015), and heart (Klotz et al, 2015; Gancz et al, 2019; Maruyama et al, 2019; Stone & Stainier, 2019; Lioux et al, 2020) were shown to form in part through lymphvasculogenic assembly of vessels from progenitors of diverse, including non-venous origins. Our results presented here from temporally restricted genetic lineage tracing using 4-OHT revealed *Cdh5*-lineage–independent origin of a significant proportion of dermal LECs. Although *Cdh5* is considered a generic marker of differentiated endothelium, our results cannot exclude the existence of a specific population of embryonic BECs that does not express *Cdh5* and/or is not targeted by the *Cdh5-CreER^T2* transgene. However, together with the previous finding that a large part of the dermal lymphatic vasculature is also not traceable using the constitutive *Tie2-Cre* (Martinez-Corral et al, 2015), the two studies provide compelling evidence for non-BEC origin of lumbar LECs. Our data do not exclude additional, including the previously described BEC-derived sources of dermal LECs (Pichol-Thievend et al, 2018), but point to their minor contribution in the lumbar skin. Importantly, a recent study identified a population of Etv2+Prox1+ lymphangioblasts that arise directly

from paraxial mesoderm-derived progenitors between E9-E10 as the primary source of initial LECs that give rise to the dermal lymphatic vasculature (Lupu et al, 2022 *Preprint*). Because these LECs do not differentiate via a venous EC intermediate, they are not expected to be traced with 4-OHT treatment at E9.5 in the *Cdh5-CreER^{T2}* line. However, additional experiments are required to confirm the lymphangioblast origin of the non–*Tie2/Cdh5* lineage–derived dermal LECs described in our study.

### Choice between tamoxifen and 4-OHT determines the temporal window for stage-specific lineage tracing

An important conclusion from our study is that the different kinetics of Cre activity induced by tamoxifen and its active metabolite 4-OHT has critical implications for the outcome of the lineage tracing experiment. The conversion of tamoxifen into its active metabolites in the liver takes ~6–12 h in vivo (Jensen & Dymecki, 2014), which is expected to result in a delay in the onset of Cre-mediated recombination when compared with administration of 4-OHT. Measurement of 4-OHT serum levels in mice administered with different doses of tamoxifen or 4-OHT indeed showed different kinetics (Zovein et al, 2008; Martinez-Corral et al, 2015) (Fig 3). Although serum levels of 4-OHT peaked after intraperitoneal 4-OHT administration at 6 h and declined within 24 h, tamoxifen administration led to slower kinetics with levels peaking at 12 h and sustained for up to 48–72 h. Repeated administration of tamoxifen was reported to induce Cre-mediated recombination for even up to weeks after the last treatment (Reinert et al, 2012; Ye et al, 2015) and is thus not compatible with stage-specific lineage tracing. An additional experimental variable affecting the efficiency of Cre-mediated recombination is the use of different commercially available 4-OHT preparations composed of different ratios of its *cis* and *trans* isoforms, of which *trans*-4-OHT exhibits over 300-fold higher affinity to estrogen receptor (ER) ligand–binding domain compared with *cis*-4-OHT (Felker et al, 2016). Because differences in 4-OHT and tamoxifen activity will directly translate into different developmental windows of Cre activity, the experimental conditions for the induction of Cre activity should be an important consideration in particular during early stages of rapid tissue differentiation.

## Materials and Methods

### Mouse lines and treatments

*Cdh5-CreER^{T2}* (Wang et al, 2010) *and Ai14* (*R26-tdTom*) (Madisen et al, 2010) mice described previously were analyzed on a C57BL/6J background. The morning of vaginal plug detection was considered as embryonic day 0 (E0). For inducing recombination during embryonic development, a single dose of 1 mg of 4-hydroxytamoxifen (4-OHT, H7904; Sigma-Aldrich, ≥98% Z [*trans*] isomer) or tamoxifen (tam, T5648; Sigma-Aldrich) dissolved in peanut oil (10 mg/ml) was administered intraperitoneally to pregnant females at a defined developmental stage as indicated in the figures and figure legends. All experimental procedures were approved by the Uppsala Animal Experiment Ethics Board (permit numbers 130/15

and 5.8.18-06383/2020) and performed in compliance with all relevant Swedish regulations.

### Antibodies

The following primary antibodies were used for whole-mount of skin: rat anti-mouse PECAM1 (Cat. no. 553370, 1:200; Becton Dickinson), goat anti-mouse NRP2 (Cat. no. AF567, 1:200; R&D Systems), and rabbit anti-PROX1 (generated against human PROX1 C-terminus [567–737 aa] [Stanczuk et al, 2015] [1:200]). Secondary antibodies conjugated to Cy3, Alexa Fluor 488 or 647 were obtained from Jackson ImmunoResearch and used in 1:300 dilution.

### Immunofluorescence

The embryonic skin was fixed in 4% paraformaldehyde at room temperature for 2 h followed by permeabilization in 0.3% Triton X-100 in PBS (PBST) for 10 min and blocking in PBST plus 3% milk for 2 h. Primary antibodies were incubated at 4°C overnight in blocking buffer and washed in PBST before incubating them with fluorescence-conjugated secondary antibodies in blocking buffer for 2 h at room temperature. Stained samples were washed with PBST and mounted in Mowiol.

### Flow cytometry

For FACS analysis of embryonic back skin, cervical and lumbar skin regions were dissected and digested separately in Collagenase IV (Life Technologies) 4 mg/ml, DNase I (Roche) 0.2 mg/ml, and FBS 10% (Life Technologies) in PBS at 37°C for about 15 min with vigorous shaking every 5 min. Collagenase activity was quenched by dilution with FACS buffer (PBS, 0.5% FBS, 2 mM EDTA), and digestion products were filtered through 70-µm nylon filters (BD Biosciences). Cells were again washed with FACS buffer and immediately processed for immunostaining first by blocking Fc receptor binding with rat anti-mouse CD16/CD32 (Cat. no. 14-0161-85, 1:100; eBioscience) followed by incubation with antibodies (all obtained from eBioscience) targeting PDPN (conjugated to AF488, clone eBio8.1.1, cat 53-5381, 1:100), CD31/PECAM1 (PE-Cy7, 390, 46-0311, 1:300), and LYVE1 (eFluor660, ALY7, 50-0443, 1:100). Immune cells and erythrocytes as well as dead cells were excluded using anti-CD45 (30-F11, eFluor450, 48-0451, 1:50), anti-CD11b (M1/70, eFluor450, 48-0112, 1:50), and anti-TER-119 (TER-119, eFluor450, 48-5921, 1:50), together with the cell death dye Sytox blue (Life Technologies), all detected by the violet laser as one dump channel. For compensation, the AbC anti-rat/hamster compensation bead kit (Life Technologies) was used. Single viable cells were gated from FSC-A/SSC-A, FSC-H/FSC-W, and SSC-H/SSC-W plots followed by exclusion of cells appearing in the violet dump channel. FMO controls were used to set up the subsequent gating scheme to obtain cell populations (Fig S1A and B).

### Image acquisition

All confocal images represent maximum intensity projections of Z-stacks of single tiles. Images were acquired using Leica SP8 confocal microscope and Leica LAS X software.

## Statistical analysis

GraphPad Prism was used for graphic representation and statistical analysis of the data. Data between two groups were compared with the paired two-tailed $t$ test, assuming equal variance. Differences were considered statistically significant when $P < 0.05$. The experiments were not randomized, and no blinding was done in the analysis and quantifications. No statistical methods were used to predetermine the sample size.

# Supplementary Information

# Acknowledgements

We thank Ralf Adams (Max Planck Institute for Molecular Biomedicine, Münster) for the *Cdh5-CreER^{T2}* mice. We also thank the BioVis facility (Uppsala University, Sweden) for flow cytometer usage and support and Sofie Lunell Segergvist, Sofie Sjöberg, and Aissatu Mami Camara for technical assistance. This work was supported by grants from Knut and Alice Wallenberg Foundation (2018.0218), the Swedish Research Council (2020-02692), the Göran Gustafsson foundation, and the Swedish Cancer Society (19 0220 Pj), all to T Mäkinen.

## Author Contributions

Y Zhang: formal analysis, validation, investigation, visualization, and writing—review and editing.
H Ortsäter: formal analysis, validation, investigation, visualization, and writing—review and editing.
I Martinez-Corral: conceptualization, formal analysis, investigation, and writing—review and editing.
T Mäkinen: conceptualization, formal analysis, supervision, funding acquisition, visualization, project administration, and writing—original draft, review, and editing.

## Conflict of Interest Statement

The authors declare that they have no conflict of interest.

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
