## [Reviewer comments · Life Science Alliance]

Life Science Alliance

Cdh5-lineage independent origin of dermal lymphatics shown by temporally restricted lineage tracing

Yan Zhang, Henrik Ortsäter, Ines Martinez-Corral, and Taija Makinen

DOI: <https://doi.org/10.26508/lisa.2022.01561>

Corresponding author(s): Taija Makinen, Uppsala University

Review Timeline:

Submission Date:	2022-06-16
Editorial Decision:	2022-07-12
Revision Received:	2022-07-21
Accepted:	2022-07-22

Transaction Report:

Please note that the manuscript was reviewed at Review Commons and these reports were taken into account in the decision-making process at Life Science Alliance.

July 12, 2022

RE: Life Science Alliance Manuscript #LSA-2022-01561

Prof. Taija Makinen
Uppsala University
Department of Immunology, Genetics and Pathology
Uppsala 751 85
Sweden

Dear Dr. Makinen,

Thank you for submitting your revised manuscript entitled "Cdh5-lineage independent origin of dermal lymphatic vessels revealed by temporally restricted lineage tracing". We would be happy to publish your paper in Life Science Alliance pending final revisions necessary to meet our formatting guidelines.

- please upload your manuscript in editable doc format
- please upload both your main and supplementary figures as single files
- please add a running title, alternate abstract, category, and Twitter handle for the author and for your institution to our system
- please consult our manuscript preparation guidelines <https://www.life-science-alliance.org/manuscript-prep> and make sure your manuscript sections are in the correct order
- please use the [10 author names, et al.] format in your references (i.e. limit the author names to the first 10)

A. FINAL FILES:

B. MANUSCRIPT ORGANIZATION AND FORMATTING:

**Submission of a paper that does not conform to Life Science Alliance guidelines will delay the acceptance of your

manuscript.**

The license to publish form must be signed before your manuscript can be sent to production. A link to the electronic license to publish form will be sent to the corresponding author only. Please take a moment to check your funder requirements.

Sincerely,

Reviewer #1 (Comments to the Authors (Required)):

The authors have sufficiently addressed my concerns from the previous version of their work. That they cite a work on Biorxiv from the authors in reference to a non-BEC origin (my last comment) is of concern as that information really should be discussed in this paper, or at least the suggested possibility, as that work cannot really be cited here. Otherwise, it is complete.

Reviewer #2 (Comments to the Authors (Required)):

All my concerns have been addressed satisfactorily. I have no further comments.

July 22, 2022

RE: Life Science Alliance Manuscript #LSA-2022-01561R

Prof. Taija Makinen
Uppsala University
Department of Immunology, Genetics and Pathology
Uppsala 751 85
Sweden

Dear Dr. Makinen,

Thank you for submitting your Research Article entitled "Cdh5-lineage independent origin of dermal lymphatics shown by temporally restricted lineage tracing". It is a pleasure to let you know that your manuscript is now accepted for publication in Life Science Alliance. Congratulations on this interesting work.

DISTRIBUTION OF MATERIALS:

Again, congratulations on a very nice paper. I hope you found the review process to be constructive and are pleased with how the manuscript was handled editorially. We look forward to future exciting submissions from your lab.

Sincerely,
